# TargetDB: A target information aggregation tool and tractability predictor

**Stephane De Cesco**◉*, **John B. Davis, Paul E. Brennan**◉*

Nuffield Department of Medicine, ARUK Oxford Drug Discovery Institute, Target Discovery Institute, University of Oxford, Oxford, United-Kingdom

* paul.brennan@ndm.ox.ac.uk (PEB); sdecesco@gmail.com (SDC)

## Abstract

When trying to identify new potential therapeutic protein targets, access to data and knowledge is increasingly important. In a field where new resources and data sources become available every day, it is crucial to be able to take a step back and look at the wider picture in order to identify potential drug targets. While this task is routinely performed by bespoke literature searches, it is often time-consuming and lacks uniformity when comparing multiple targets at one time. To address this challenge, we developed TargetDB, a tool that aggregates public information available on given target(s) (links to disease, safety, 3D structures, ligandability, novelty, etc.) and assembles it in an easy to read output ready for the researcher to analyze. In addition, we developed a target scoring system based on the desirable attributes of good therapeutic targets and machine learning classification system to categorize novel targets as having promising or challenging tractability. In this manuscript, we present the methodology used to develop TargetDB as well as test cases.

## Introduction

With the rising availability of genome-wide association data (GWAS) [1], proteomics [2,3], CRISPR [4–6] and RNAi [7], the list of potential protein targets for a given disease is growing rapidly. In this context, researchers are spoilt for choice when it comes to picking a target for further investigation, and yet the failure rate in clinical trials suggests that researchers are routinely failing to select the best targets against which to pitch their drug discovery efforts. To help them in this task, a plethora of excellent publicly available resources exist, such as UniProt [8], DrugBank [9], ChEMBL [10], Open Targets [11], Therapeutic Target Database (TTD) [12], The Drug Gene Interaction database (DGIdb) [13], Target Central Resource Database (TCRD) [14] and many others [15]. While they all provide valuable information, combining all this information in a single place for further analysis or prioritization of a list of targets can become a daunting task. With each data source specializing in different areas such as protein expression, disease association or pharmacology, researchers are required to collate and navigate through a miriad of cross-references in order to paint an accurate portrait of a potential target. Although resources such as UniProt, Pharos/TCRD and Open Targets already propose some aggregation of data, we propose with TargetDB to complement them with additional

**Data Availability Statement:** All of the data used and code developed for this project is deposited in a GitHub repository (https://github.com/sdecesco/targetDB)

**Funding:** This work was supported by Alzheimer's Research UK [registered charity 1077089 and SC042474].

**Competing interests:** No competing interest to declare.

information such as structurally enabled druggability assessment, area-specific scoring for agile prioritization and a tractability prediction model. More recently, a tool with similar features, TractaViewer, has been described in the literature [16]. While this tool allows the user to classify targets into different bins, it does not provide an area specific score or a general scoring system that can be used for target prioritization. The bin assignment combined with the scoring system of TargetDB, however, could provide valuable information to researchers seeking to assess target tractability.

## Materials and methods

### TargetDB

TargetDB is distributed as a python package and a pre-built SQLite database. The user can also build the database from scratch using a command-line interface in Linux based systems. Details on the database and on how to install the package are available in the Supplementary Information (S1 File) and on the GitHub page (https://github.com/sdecesco/targetDB).

### Data sources

Data used in TargetDB comes from a variety of sources. Some data comes from pre-aggregated/processed data from other databases such as UniProt or TCRD. While others come directly from the source API's such as Human Protein Atlas for protein expression levels and Open Targets for disease association. The full list of data sources is available in the Supplementary Information (S1 File).

### Structural assessment of druggability

Fpocket [17] (version 3) was used in order to probe the potential ligandability of queried targets by assessing the presence of protein pockets amenable for small molecule binding (https://github.com/Discngine/fpocket). For each target in the database, PDB files were downloaded locally and only the smallest biological assembly with a chain representing the target of interest was kept for further analysis. Fpocket was then used with the default parameters and output files read and incorporated into the TargetDB database.

### Tractability model

Data collated in TargetDB is then retrieved and used to generate a series of descriptors that are used for: 1) calculate the area-specific overall score, 2) as input for machine learning algorithm in order to predict the target tractability. The final model uses the random forest algorithm from the python package sci-kit-learn [18]. The building of the model is discussed in the results and detailed procedures and code, in the form of a jupyter notebook, and training/testing data are available in the GitHub repository.

## Results

Once the program and database are downloaded, TargetDB can be run as a Tkinter graphical interface where different modes can be selected (Fig 1): Single Mode, List Mode and Spider Plot mode. For each mode, the target(s) of interest need to be specified. In Single Mode, one file is generated per gene entered and, while nothing prevents the user from using this mode for a large number of targets, it is best suited for a handful of genes. For a large number of targets the List Mode is more appropriate, as it produces a single file with several columns that allow the user to prioritize targets according to many attributes. In Spider Plot mode, a graphical spider plot representation of a single target landscape is depicted, representing the amount

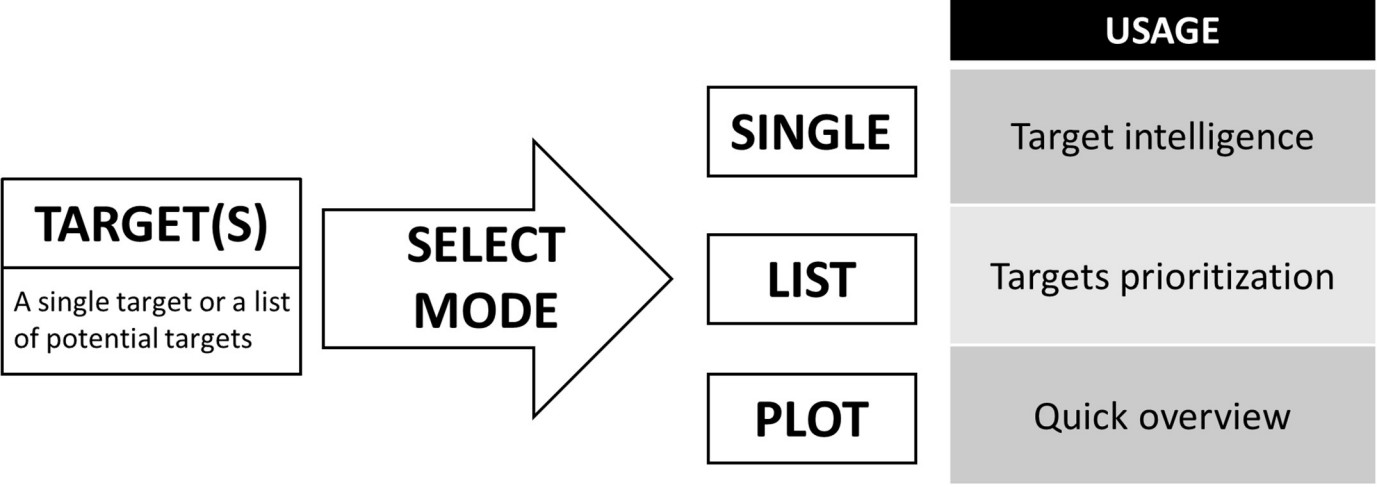

**Fig 1. TargetDB modes.** These are the available worklflows and recommended usage for the TargetDB output formats.

of knowledge on a target in different areas. This plot is also included in the Single Mode output. An example of each output is available in the supporting information.

## Aggregated information about a specific target

The excel document (S2 File) generated from the database contains several worksheets with different information regarding the target. The main page contains general information as well as the spider plot. Detailed sheets provided are listed below with a short description.

**Pubmed search.** A pubmed search using the gene name as search term is conducted and the 500 most recent publications are listed in the worksheet.

**Diseases.** This worksheet contains the protein expression (upregulated or downregulated) and GWAS associations for the target in the context of different diseases. This data comes from the Humanmine datasource [19].

**Open target association.** Disease associations come from the Open Targets platform. The individual disease, disease areas and association type scores are displayed.

**Expression.** Protein expression levels come from the Human Protein Atlas portal. Numerical values can be interpreted as the following: 3 = High level of expression; 2 = Medium level of expression; 1 = Low level of expression; 0 = Not observed.

**Genotypes.** List of different mouse genotypes (Knockout, knockdown, etc.) for the target of interest with their associated observed phenotypes. Green color identifies genotypes with no abnormal phenotypes observed, while red indicates a genotype with a lethal phenotype observed.

**Isoforms.** List of different isoforms with their associated sequence differences.

**Variants/Mutants.** List of observed variants and mutants along with their sequences and the effect observed if available.

**Structure.** This worksheet contains a list of all available structures available in the PDB. The code, along with the technique, resolution, chain and sequence coverage, is listed together with information from PDBBind. On top of that, details on domains and their tractability/druggability coming from DrugEbillity is also displayed.

**Pockets.** After analysis of potential small molecule binding pockets with fpocket3, the results are imported into TargetDB and are displayed in this sheet. The ligandability score is generated directly by the fpocket3 algorithm and we refer the reader to the original paper for

more details about the method used to generate this score [20]. As a general guideline, a ligandability binding pocket will have a score of over 0.5, up to a maximum of 1. If multiple pockets are found for a single structure, a complete list of them will be output. If no druggable pocket is found in the target PDB or no PDB is available for the target, a BLAST search is performed on sequences that have a crystal structure deposited in the PDB. A similar pocket analysis is then performed, and the result displayed in the output document with the identified target as well as the sequence similarity between them.

**Binding.** Bioactivities extracted from ChEMBL contain many different types of data and, while they all provide valuable information, it was decided to segregate the data into different sheets: Binding, Dose-response, Percent Inhibition, ADME and Other bioactivities. The Binding sheet only contains Ki/Kd datapoints. Bioactivities of a given ligand against other targets were collected and used to calculate a selectivity score (Selectivity Entropy—Shannon Entropy [21]), the name of the target for which the ligand has the best bioactivity is also displayed. To provide more information about the ligands, physicochemical properties, as well as the CNS MPO [22] score, are also provided.

**Dose-response/Percent-Inhibition/ADME/Other bioactivities.** Similar to the above mentioned but with different data types.

**BindingDB/Commercial compounds.** Similar to the above mentioned with BindingDB as the datasource. The commercial compounds worksheet contains a link to the chemical suppliers of BindingDB ligands for the target.

## Prioritize a list of candidate targets

The target List Mode report provides the user with more than a hundred different metrics to define a potential target (S3 File) such as: number of crystal structures in the PDB, ChEMBL bioactive compounds, Open Targets disease associations, number of antibodies, human protein expression levels in tissues, etc. Such an abundance of available fields makes it hard to quickly identify a target's profile or else to pick the most relevant parameters for the prioritization process. Therefore, we use a set of rules to define area-specific scores that aid target assessment and prioritization.

**Area-specific scores for rapid target assessment.** When evaluating potential targets, building an overall picture of a target profile is not an easy task with the information often fragmented across numerous resources. With TargetDB we have separated information into eight main categories: Druggability, Structure, Biology, Chemistry, Diseases, Genetics, Information and Safety (Fig 2). Each category is scored from zero to one according to a set of rules (S1 File). Once calculated these scores can be used to generate a spider plot of the target profile to rapidly identify the strengths and weaknesses of a given target in each category. From the few examples in Fig 3, it is easy to identify all these targets are well studied and associated with diseases (neurodegeneration), although only some of these have genetic evidence to support the observation. While acetylcholine esterase and beta-secretase 1 are highly druggable and drugged, it is interesting to note that APOE, one of the main risk factors for Alzheimer's disease [23], does not score well in the druggability and chemistry area, which is consistent with the poor druggability of an apolipoprotein. These well-characterized examples illustrate how this representation allows for a quick interpretation of a target landscape. A guide for the interpretation of these spider plots is available in S1 Fig.

**Multi-Parameter Optimization (MPO) score for target ranking.** While ranking targets based on their area score could be used on its own, we also incorporated a customizable MPO score to allow multiple interpretations of the same data. For example, depending on the user interest for a structurally enabled target, it may be advantageous to prioritize targets for which

| Druggability | Structure | Chemistry | Biology |
|---|---|---|---|
| • Pockets druggability<br>• Domain druggability<br>• Druggability of similar targets' pockets | • % of sequence covered<br>• % of domains covered<br>• Number of PDB<br>• Number of PDB of similar targets | • BindingDB potent ligands<br>• BindingDB ligands in phase 2 clinic<br>• ChEMBL potent ligands<br>• ChEMBL selective ligands<br>• Commercial ligands | • Protein expression levels<br>• Number of antibodies<br>• Variants<br>• Mutants<br>• Mice genotypes<br>• KEGG/Reactome |
| Disease Link | Genetic links | Information | Safety |
| • Number of disease areas<br>• Max association score<br>• Diseases count | • GWAS association count<br>• Genetic association score | • JensenLab Pubmed score | • Heart protein expression<br>• Liver protein expression<br>• Kidney protein expression<br>• Lethal phenotypes observed in mice |

**Fig 2. Area-specific scores.** Different features that were selected for the generation of the area-specific scores.

3D structures are available and with a high druggability score. By simply adjusting the weightings of each category, one can generate a tailored MPO score to facilitate prioritization according to key criteria (Fig 4). Likewise, negative weights can be set to deprioritize high ranking targets and prioritize low ranking targets, this can be useful if, for example, one wants to deprioritize targets for which there is already significant chemical matter available. The decision on how to set the different weights relies on user judgment and the specific criteria that are of interest to them. As a note of caution, while a user might decide to over-prioritize areas such as structural biology and chemistry it is by no means a guarantee that this will lead to a target with potential therapeutic applications. On the other hand, if a target is over-prioritized for strong disease and genetic links, it may be very difficult to develop safe and effective therapeutics due to low safety and structural druggability scores. This facility also enables users to tailor their searches to highlight targets where there is an opportunity for their research expertise to make a larger impact by picking targets for which there is a lack structural information, lack of chemistry or lack of biology, for example. The detailed methodology on how this MPO score is calculated is available in S1 File.

**Target tractability model.** To further assist the decision-making process on target tractability, it was decided to evaluate whether or not a model of tractability could be built. With the vast amount of information collected, we might uncover trends that would allow classification of targets into tractability classes. To do so, several machine learning algorithms were tested, and their performance evaluated to predict target tractability. In order to train and evaluate the different models we needed to provide each algorithm with an annotated set of tractable and intractable targets. While finding a list of tractable targets is relatively easy, identifying a list of intractable targets proved to be more challenging. We used the DGIdb [13] "clinically actionable" annotated genes as our tractable list of targets (n = 399), while for the intractable control we selected a random set of targets (n = 400) from the list of targets present in

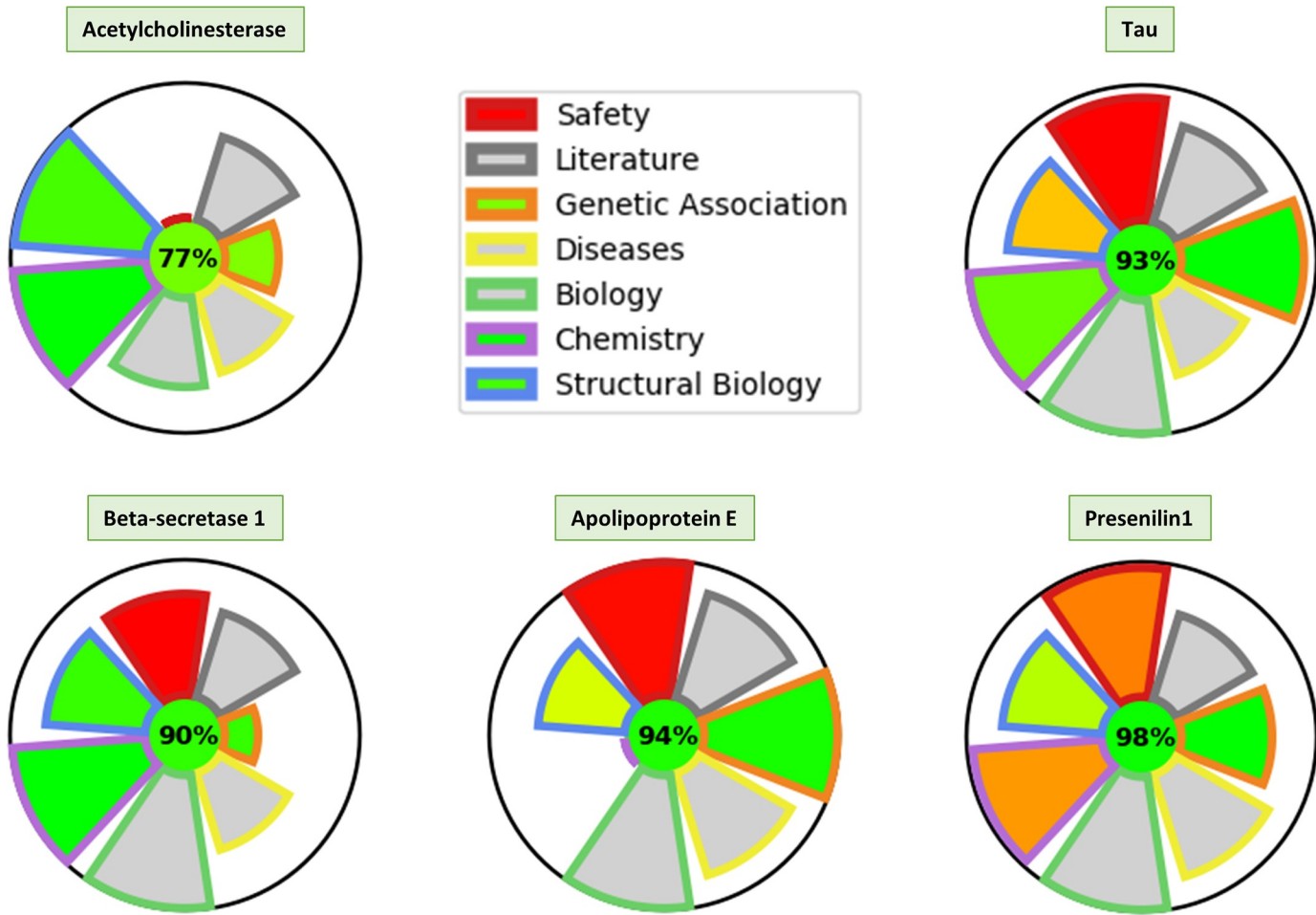

**Fig 3. Spider plot for various targets.** The Height of each section represents the amount of information available in that area for this target. The color in the safety, genetic association, chemistry and structural biology are indications of the safety risk, the significance of associations, the quality of the chemical matter and the druggability of potential binding pockets respectively (Green = better quality/safety Red = poor quality/safety risk).

TargetDB from which were removed the clinically actionable (n = 399) and the druggable genome (n = 6106) list from DGIdb. This combined set was then split into a training set (n = 560) and testing set (n = 240), each containing the same ratio of tractable/intractable targets. The complete list of targets used in the training and testing of the model is available in the S4 File.

After evaluation of multiple algorithms (a detailed procedure is available in S1 File as well as the Jupyter notebook in S5 File), the Random Forest algorithm was selected for further optimization (Fig 5). This method provides multiple advantages, such as reducing overfitting, the ability to extract information on features contributing to the decision, and providing an estimate of the confidence of the prediction. This allowed us to narrow down to a set of features that were truly contributing to the performance of the model. The underlying concept of this method is simple: the algorithm creates multiple decision trees; for each decision tree it selects a subset of features from the entire set available; all the decisions from all the trees are then compiled and a classification based on consensus is made for each target. After feature and parameter optimization, the model was evaluated against the test set and was able to accurately predict target tractability 85% of the time. A detailed description of the model performance is

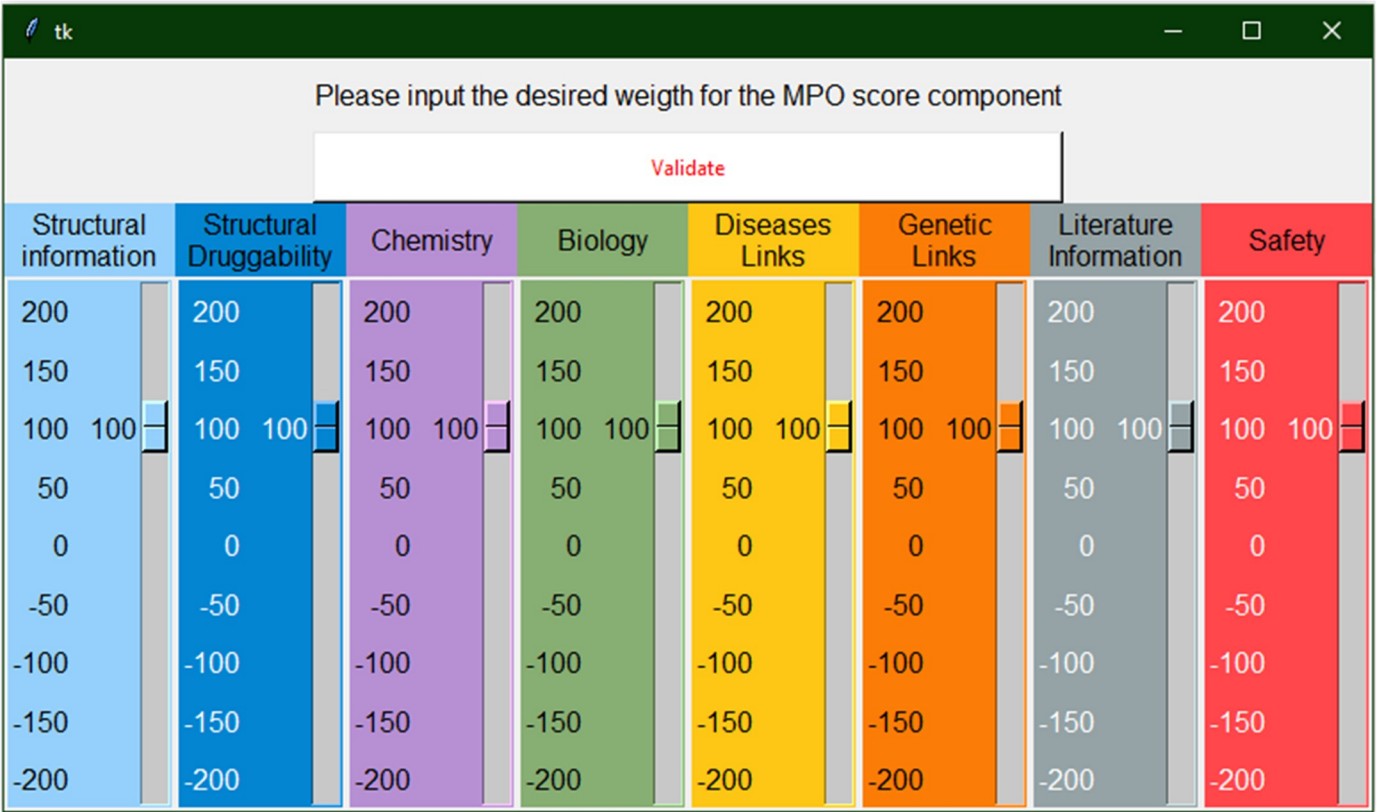

**Fig 4. MPO weight input panel.** Interface to enter the individual weights for the construction of the MPO score.

available in S1 File. The output of this model then provides two (related) readouts: Percentage of trees predicting the target to be tractable and the tractability class of a target (Tractable [>60%] – Challenging [60%-40%] – Intractable [<40%]).

## Discussion

To showcase the application of such a tool, we present here a workflow that was used to prioritize potential targets from a list of genes involved in Alzheimer's diseases provided by the AMP-AD consortium (https://agora.ampadportal.org). This list consists of 95 targets generated by 6 different teams using computational analysis of genomic, proteomic and/or metabolomic data from human samples [24,25]. Manual aggregation and collation of information for 95 targets is a time-consuming task but the same results can be achieved in only a few minutes using TargetDB. Once the program is started the user has only to input the list of targets in the window and select the run mode (single, list, plot); in our case the "List Mode" was selected. Once started, the program will take a few minutes to retrieve all the information in the database and another window will open to allow the user to input each area weight necessary to calculate a custom MPO score. In our case, the following weights were used: Structural information (= 100), Structural Druggability (= 150), Chemistry (= -100), Biology (= 100), Diseases Links (= 100), Genetic Links (= 150), Literature information (= -100), Safety (= 0). The rationale is that we want to select structurally druggable targets, with no or little chemistry available and with strong genetic associations. Biological information and link to diseases are parameters to consider but not essential and we wanted to deprioritize targets with large amounts of

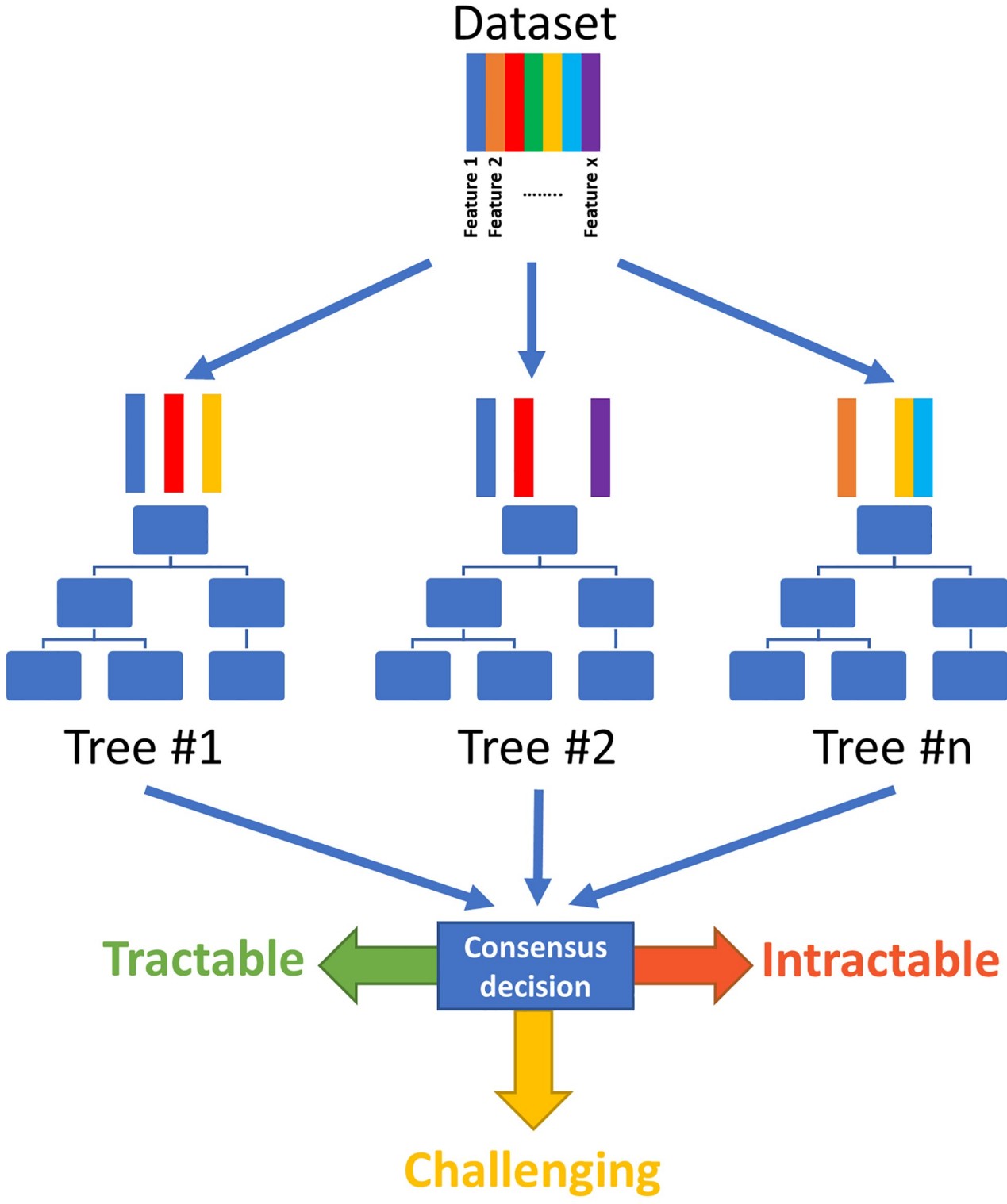

**Fig 5. Random forest.** Principle behind the Random Forest Machine learning algorithm.

**Table 1. Different MPO scenario.**

| SBDD MPO | | Crystallography MPO | |
| --- | --- | --- | --- |
| Gene Name | MPO Score | Gene name | MPO Score |
| GRIN2A | 0.8 | SGPL1 | 0.73 |
| PLEC | 0.79 | ALK | 0.69 |
| TGFBR2 | 0.78 | SYNGAP1 | 0.68 |
| **PLCG2** | 0.78 | S1PR1 | 0.67 |
| CFH | 0.78 | NEFL | 0.67 |
| TGFB1 | 0.76 | CSF1R | 0.65 |
| AP2B1 | 0.76 | **PLCG2** | 0.63 |
| MSN | 0.74 | NR1H4 | 0.62 |
| ERBB3 | 0.74 | GFAP | 0.62 |
| TREM2 | 0.73 | PPARA | 0.62 |

Comparison of the top10 ranked target for two different MPO Score scenarios. Left: Structure based drug design (SBDD). Right: Crystallography.

literature available. In this instance, the weight for the safety term was set to zero and was not, therefore, considered in the MPO scoring. Once the weights were entered, the program generates an excel spreadsheet with the list of targets and the calculated area-specific, tractability prediction and MPO scores (S6 and S7 Files). This spreadsheet can then be used to further refine the selection according to the user's preferences.

The same list was independently examined by scientists for target selection. 4 targets were selected for further target validation work and early hit identification. When compared to TargetDB output ranking, 3 of these 4 targets were ranked in the top 10. Assessing these 95 targets took in total a few months and several meetings; it is a good example of how TargetDB may be used to accelerate and focus the attention onto the most promising targets, while not completely discarding the lesser ranked targets for further exploration.

Interestingly, other MPO criteria can be selected depending on the kind of work that is envisioned. For example, a team mainly interested in solving crystal structures might deprioritize targets with a solved crystal structure (Structural information < 0) but still possessing favourable druggability potential calculated from data for close analogs (Structural Druggability ≥ 100) and with some therapeutic rationale (Genetic Links, Disease Links ≥100). These different criteria lead to a ranking significantly different to the first one (Table 1).

Another application is the prioritization of an entire family of proteins. We showcase here how TargetDB was used to rapidly provide an overview of the solute carrier (SLC) family of transporters (Fig 6 and S8 File). In less than an hour, we were able to shortlist potential targets based on their predicted tractability class, their MPO score, but also on the potential association with a disease of interest (Alzheimer's (AD) or Parkinson's disease (PD) in this case). After further assessment of the top targets, several of them are now under investigation within our institute. This case illustrates how TargetDB can be usefully inserted into the target discovery workflow to expedite as well as standardize the target prioritization process.

## Conclusion

In conclusion, we present a tool that allows a researcher to extract/combine and standardize outputs from many different publically accessible databases and to rapidly compare the potential of multiple targets. TargetDB is freely available as a python package and detailed installation instructions are available on the project's GitHub page as well as in the supporting

## Solute Carrier transporters (SLC)

**415**

**Disease link**

**AD** 19 · 5 2 12

**PD** 16 · 6 2 8

**Literature**

**Tractable** 48 11%

**Challenging** 61 15%

**Intractable** 306 74%

**>20%** In Dark Proteome

**106** targets with dementia linked literature

**>90%** without structure

**15%** of which with structure of close analogue

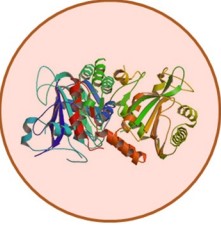

**Fig 6. Target class analysis.** Summary of the analysis of the Solute Carrier (SLCs) family of transporters performed using TargetDB in list mode as drug targets for Alzheimer's disease (AD) or Parkinson's disease (PD). The definition of dark proteome is taken from the Target Central Resource Database [26].

information. While further improvements and additions are already being considered we encourage other users to participate in the project by adding their own additional datasources.

## Supporting information

**S1 File. Additional experimental section.** Details of the different methods, datasources and calculations performed for the creation of TargetDB, discussion on the machine learning model.
(DOCX)

**S2 File. Example of single mode output.** Output from the single mode for the gene BACE1.
(XLSX)

**S3 File. Description of all the list mode columns.**
(XLSX)

**S4 File. Genes used in the machine learning model.** Excel file with the ID of the genes used in the machine learning model training and testing.
(XLSX)

**S5 File. Jupyter notebook and training data.** Archive containing the jupyter notebook used to generate the tractability model.
(ZIP)

**S6 File. AMP-AD nominated list—Medicinal chemistry ranking.**
(XLSX)

**S7 File. AMP-AD nominated list—Structural biology ranking.**
(XLSX)

**S8 File. Solute Carrier Protein (SLC) prioritization list.**
(XLSX)

**S1 Fig. Guide to interpretation of spider plots.**
(PNG)

## Acknowledgments

We would like to thank the AMP-AD consortium for providing the target list that was evaluated and the entire Oxford Drug Discovery Institute team for the effort put into evaluating these targets as well as providing useful information for the creation of this tool.

## Author Contributions

**Conceptualization:** Stephane De Cesco, John B. Davis, Paul E. Brennan.

**Formal analysis:** Stephane De Cesco.

**Methodology:** Stephane De Cesco, John B. Davis, Paul E. Brennan.

**Project administration:** Paul E. Brennan.

**Resources:** Paul E. Brennan.

**Software:** Stephane De Cesco.

**Supervision:** John B. Davis, Paul E. Brennan.

**Writing – original draft:** Stephane De Cesco.

**Writing – review & editing:** Stephane De Cesco, John B. Davis, Paul E. Brennan.

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
