## [Decision Letter · Decision Letter 0]

5 Jun 2020

PONE-D-20-11085

TargetDB: A target information aggregation tool and tractability predictor

PLOS ONE

Dear Dr. De Cesco,

Thank you for submitting your manuscript to PLOS ONE. After careful consideration, we feel that it has merit but does not fully meet PLOS ONE’s publication criteria as it currently stands. Therefore, we invite you to submit a revised version of the manuscript that addresses the points raised during the review process.

As you can see the 3 reviewers were quite positive overall, and all recommend publication. Please pay attention to the comments and respond accordingly---I would be happy to accept a revised version of this manuscript as soon as I receive and review your modified paper. 

We look forward to receiving your revised manuscript.

Kind regards,

Joseph J Barchi

Academic Editor

PLOS ONE

Journal Requirements:

2. Please ensure that you refer to Figure 2 in your text as, if accepted, production will need this reference to link the reader to the figure.

3. Please upload a copy of Figure 7, to which you refer in your text. If the figure is no longer to be included as part of the submission please remove all reference to it within the text.

Additional Editor Comments (if provided):

Reviewers' comments:

Reviewer's Responses to Questions

**Comments to the Author**

1. Is the manuscript technically sound, and do the data support the conclusions?

Reviewer #1: Yes

Reviewer #2: Yes

Reviewer #3: Yes

2. Has the statistical analysis been performed appropriately and rigorously? 

Reviewer #1: Yes

Reviewer #2: I Don't Know

Reviewer #3: N/A

3. Have the authors made all data underlying the findings in their manuscript fully available?

Reviewer #1: Yes

Reviewer #2: Yes

Reviewer #3: Yes

4. Is the manuscript presented in an intelligible fashion and written in standard English?

Reviewer #1: Yes

Reviewer #2: Yes

Reviewer #3: Yes

5. Review Comments to the Author

Reviewer #1: The authors have developed a handy tool to assist researchers with the challenging task of prioritising a list of potential targets. A large number of various data resources are integrated and final outputs in terms of excel sheets are reported. The ability to adjust weights for main categories is particular useful to customise ranking of targets.

1.) We installed the targetDB package via conda as described on the github page, on MacOS. An initial run failed due to missing xlsxwriter and xlrd packages. Perhaps those dependencies can be mentioned as part of the installation instructions. After installation of those packages the software ran successfully.

2.) Occasionally, an error is thrown, presumably if no protein expression levels are available.

KeyError: "['Muscle tissues', 'Female tissues', 'Bone marrow & lymphoid tissues', 'Liver & gallbladder', 'Proximal digestive tract', 'Gastrointestinal tract', 'Kidney & urinary bladder', 'Male tissues', 'Adipose & soft tissue', 'Endocrine tissues'] not in index"

For example, when entering SLC6A8 and SLC6A5 as the gene list in 'List mode'. These corner cases should be caught and dealt with instead of left just hanging.

3.) There is a spreadsheet called "Not in DB" in List mode. Is this supposed to provide the user with the subset of genes that were not found/matched in the DB? This is useful if dozens of genes were entered and the user can quickly check which genes were not found. However, the spreadsheet is empty at the moment if gene names were not found.

4.) The mode 'Spider plot only' does not give an option to save the produced figure, at least not on MacOS.

5.) A supplementary table should be included listing the 399 tractable targets, 400 intractable targets and their allocation into training and test sets.

6.) Table 1, it is not clear what 'SBDD' stands for.

7.) Figure 6, provide the reader with a short explanation of the 'dark proteome'

8.) What about other tools with very similar functionality, e.g. differences between TargetDB and the recently published TractaViewer should be discussed.

9.) It's somewhat unfortunate that the name "TargetDB" has been used previously for another DB (PMID: 15130928). The authors might want to consider renaming their software/DB, although it seems that this older DB and its original name is not in use anymore.

Reviewer #2: This paper presents a useful tool that aggregates information on a given human gene and presents that information in an easy to read and manipulate format. I can see that this tool would be very useful for prioritising gene targets in the context of early stage drug discovery in industry and also academic pipelines. The ability to up/down-weight the various parameter spaces in terms of the scoring is a nice feature which makes this a generically useful tool.

I was able to download and install the program successfully and run a number of targets through it. I only tested under Linux so cannot say if it would work well under Windows or Mac.

Some comments and requests:

• I note that the supplied database is for ChEMBL 24, which is now quite out of date. I would like to see some instructions on how a newer version of ChEMBL (i.e. 26) can be used, notwithstanding the fact that it’s likely some code changes will be required to accommodate any ChEMBL schema changes

• I would like to see guidance on how the program can be extended in principle. For example, population information on common SNPs, or integration of data from gnomAD.

• The spider plots are very useful visualisations. However, it’s a bit hard to get your head around the filled colour schemes in the single gene output. I would like to see a legend underneath to explain the fill colours to the reader.

• I would like to see a more detailed discussion on the machine learning process and outcomes in the text. Whilst the jupyter notebook is reasonably understandable, more discussion is required on why the exact form of Random Forest was chosen, in the context of the other methods tested. This is important because it gives the community a better understanding of your experience of using a large and diverse parameter space, which others can take advantage of.

• You state that “Safety is not considered in the MPO scoring at this time”. It is not clear to me whether you mean that the MPO weighting is not considered or just that in the example you set this to zero. Please clarify.

• Open Targets is ‘misspelled’ in a number of places. Please use the correct version ‘Open Targets’ consistently.

• Please provide a reference to Humanmine.

• Please can you describe in more detail what the selectivity score is? You specific that it is (selectivity entropy – Shannon entropy) and reference the seminar 1948 information theory paper, but there is not enough information here to reproduce what you are trying to achieve.

• It appears that there are no resolution cut-offs used for structures that go through fpocket analysis. Please can you explain if that is the case and why you feel this is appropriate, if so? An EM structure of >6 Angstroms is going to provide unreliable results in this context, for example.

• What is the sustainability plan for targetDB? Since databases change all the time, how will you ensure that this platform is still usable even in 6 months’ time?

• It is highly unusual to use ellipses in articles. Please remove these and use ‘etc.’ (for example) instead.

• It would be good to have the article carefully proof read for English before it is finally accepted.

Overall, a nice piece of work that I’m sure many will want to use.

Reviewer #3: The authors construct a database application with a graphical interface that aggregates multiple disparate sources of evidence for novel drug target evaluation. They consider measures of druggability, structure, chemistry, biology, disease association, genetic association, general information, and safety when constructing their application. As output they produce different summaries of this evidence - either a summary for a single target, multiple targets, or a graphical summary (aka a spider plot). Furthermore, the authors provide an optimization approach (powered by a machine learning method) to allow users to weight multiple classes of evidence based on the users target validation needs and interests.

Major comments -

Overall the authors provide a potentially useful tool to aid structural biologists, chemists, and assay developers for target prioritization. That being said, I have the following comments:

1. How reproducible are the results generated by this - many of the databases that evidence is being pulled from may change over time. Is there some way to version the results, or to construct queries that refer to a specific version of a public database?

2. The user defined weighting is an interesting idea - I would like to see the authors discuss potential pitfalls of this - e.g. over optimizing for targets with interesting structural biology/chemistry as opposed to targets that will actually lead to therapies or deeper insights into the disease biology.

Minor comments -

1. use of ellipses is distracting (Abstract, Introduction Paragraph 1).

2. “Percentage of threes predicting” - spelling/grammar throughout should be double checked.

3. SBDD acronym in Table 1 is not defined.

6. PLOS authors have the option to publish the peer review history of their article (what does this mean?). If published, this will include your full peer review and any attached files.

Reviewer #1: No

Reviewer #2: No

Reviewer #3: No

---

## [Author Response · Author response to Decision Letter 0]

22 Jul 2020

PREFACE: Some answers to reviewers contains figures or equations, therefore, all answers have been added to the cover letter as well

Reviewers’ comments to the authors:

Reviewer #1

The authors have developed a handy tool to assist researchers with the challenging task of prioritising a list of potential targets. A large number of various data resources are integrated and final outputs in terms of excel sheets are reported. The ability to adjust weights for main categories is particular useful to customise ranking of targets.

Answer: We thank the reviewer for their kind words and are happy that he/she finds it useful.

1.) We installed the targetDB package via conda as described on the github page, on MacOS. An initial run failed due to missing xlsxwriter and xlrd packages. Perhaps those dependencies can be mentioned as part of the installation instructions. After installation of those packages the software ran successfully.

Answer: Thanks for the comment, the conda package wasn’t yet created at the time of submission and it hasn’t been tested extensively. As it is another contributor that created the package I asked him to include these packages to the dependency list. 

2.) Occasionally, an error is thrown, presumably if no protein expression levels are available.

KeyError: "['Muscle tissues', 'Female tissues', 'Bone marrow & lymphoid tissues', 'Liver & gallbladder', 'Proximal digestive tract', 'Gastrointestinal tract', 'Kidney & urinary bladder', 'Male tissues', 'Adipose & soft tissue', 'Endocrine tissues'] not in index"

For example, when entering SLC6A8 and SLC6A5 as the gene list in 'List mode'. These corner cases should be caught and dealt with instead of left just hanging.

Answer: It has now been fixed in the code in version 1.3.1 of the python package

3.) There is a spreadsheet called "Not in DB" in List mode. Is this supposed to provide the user with the subset of genes that were not found/matched in the DB? This is useful if dozens of genes were entered and the user can quickly check which genes were not found. However, the spreadsheet is empty at the moment if gene names were not found.

Answer: It has now been fixed in the code in version 1.3.1 of the python package

4.) The mode 'Spider plot only' does not give an option to save the produced figure, at least not on MacOS.

Answer: We thank the reviewer for the suggestion, it has now been added to the version 1.3.1 of the python package. 

5.) A supplementary table should be included listing the 399 tractable targets, 400 intractable targets and their allocation into training and test sets.

Answer: An additional SI document titled “ML_TrainingTestingSplit_Targets.xlsx” has been added. The document contains two tabs with the testing and training set and each has 2 columns: Target Uniprot ID and Druggability class (1 = Druggable, 0=Intractable) 

6.) Table 1, it is not clear what 'SBDD' stands for.

Answer: This has now been clarified in the legend of the table. 

7.) Figure 6, provide the reader with a short explanation of the 'dark proteome'

Answer: We have added the following text under the figure: “The definition of dark proteome is taken from the Target Central Resource Database.” Including a reference to the definition on the TCRD platform website ( http://juniper.health.unm.edu/tcrd/)

8.) What about other tools with very similar functionality, e.g. differences between TargetDB and the recently published TractaViewer should be discussed.

Answer: A sentence has been added to discuss the differences between the two tools:

“More recently, a tool with similar features, TractaViewer, has been described in the literature [16]. While this tool allows the user to classify targets into different bins, it does not provide an area specific score or a general scoring system that can be used for target prioritization. The bin assignment combined with the scoring system of TargetDB, however, could provide valuable information to researchers seeking to assess target tractability

9.) It's somewhat unfortunate that the name "TargetDB" has been used previously for another DB (PMID: 15130928). The authors might want to consider renaming their software/DB, although it seems that this older DB and its original name is not in use anymore.

Answer: While we certainly agree with the reviewer, we eventually decided not to change the name as, the reviewer pointed out, the older platform has not been in use for a long time. We believe the TargetDB name reflects best the purpose of this tool. Furthermore, a significant number of academic research teams in the UK and a few pharmaceutical companies are already familiar with the TargetDB name and are using our platform. Therefore, we feel it would be better to continue to use TargetDB

Reviewer #2

This paper presents a useful tool that aggregates information on a given human gene and presents that information in an easy to read and manipulate format. I can see that this tool would be very useful for prioritising gene targets in the context of early stage drug discovery in industry and also academic pipelines. The ability to up/down-weight the various parameter spaces in terms of the scoring is a nice feature which makes this a generically useful tool.

Answer: We thank the reviewer for their kind words and are happy that he/she finds it useful.

I was able to download and install the program successfully and run a number of targets through it. I only tested under Linux so cannot say if it would work well under Windows or Mac.

Some comments and requests:

• I note that the supplied database is for ChEMBL 24, which is now quite out of date. I would like to see some instructions on how a newer version of ChEMBL (i.e. 26) can be used, notwithstanding the fact that it’s likely some code changes will be required to accommodate any ChEMBL schema changes

Answer: The readme file on github has been updated to reflect the fact that the data in the present release has been used with ChEMBL 25. The next update of the database (scheduled for August) will be using ChEMBL 27 or the most recent release at the time of the update.

• I would like to see guidance on how the program can be extended in principle. For example, population information on common SNPs, or integration of data from gnomAD.

Answer: All the source code is available on GitHub and anyone with experience in python can branch the repository to add any valuable information to the output. This may be completed in a couple of steps: 

- Convert/parse the data into a pandas dataframe (example of this is in the druggability_DB.py file) and store it as an attribute of the Class Target (line 1290)

- Add this dataframe to the sqlite database (and create a new table) using the function write_to_db() (line 1074 - druggability_DB.py)

- To add it to the output these three files would need to be updated accordingly as well: 

 targetDB\\druggability_report.py

 targetDB\\target_descriptors.py

 targetDB\\target_features.py

We have added a sentence to convey this message of collaborative effort more clearly:

“We strongly encourage anyone interested to download the code and participate in the project by adding new datasources and/or features.”

• The spider plots are very useful visualisations. However, it’s a bit hard to get your head around the filled colour schemes in the single gene output. I would like to see a legend underneath to explain the fill colours to the reader.

Answer: We agree that the spider plot combines a lot of data in different forms and a reading guide would be beneficial. We have added a key to the SI (S1_Fig), as well as to the documentation on the github page of the project, to help the user interpret the plot. We hope the reviewers find this useful.

• I would like to see a more detailed discussion on the machine learning process and outcomes in the text. Whilst the jupyter notebook is reasonably understandable, more discussion is required on why the exact form of Random Forest was chosen, in the context of the other methods tested. This is important because it gives the community a better understanding of your experience of using a large and diverse parameter space, which others can take advantage of.

Answer: While we certainly agree with the reviewer that others can take advantage of the process we went through, we believe that the technical details of the model training and evaluation are better suited for the supplemental information (S1 File), where we go into detail about picking/removing collinear features and selection/refinement of the best algorithm. We have added the following sentence to the main text to provide better context to why the random forest algorithm was picked: 

“This allowed us to narrow down to a set of features that were truly contributing to the performance of the model.”

• You state that “Safety is not considered in the MPO scoring at this time”. It is not clear to me whether you mean that the MPO weighting is not considered or just that in the example you set this to zero. Please clarify.

Answer: We are sorry this was not more clear. The original meaning was that the weight was set to zero in that specific instance. We have now reformulated the sentence to make it clearer. The sentence now reads: 

“ In this instance, the weight for the safety term was set to zero and was not, therefore, considered in the MPO scoring”

• Open Targets is ‘misspelled’ in a number of places. Please use the correct version ‘Open Targets’ consistently.

Answer: This has now been fixed in the revised manuscript.

• Please provide a reference to Humanmine.

Answer: We are sorry to have missed that, it has been added to the revised manuscript

• Please can you describe in more detail what the selectivity score is? You specific that it is (selectivity entropy – Shannon entropy) and reference the seminar 1948 information theory paper, but there is not enough information here to reproduce what you are trying to achieve.

Answer: A more detailed calculation has been added in the Supporting information S1 File

The Shannon entropy was used as a measure of selectivity for the two following metrics. The global equation is the following:

S_sel=-∑_i^T▒〖ρ_i log⁡〖ρ_i 〗 〗

With Ssel = Selectivity Entropy

Selectivity of compound

 T = Kd for different targets 

 ρ_i = Probability for a Kd value

ρ_((T) )=〖K_d〗_T /(∑_i▒〖K_d〗_i )

 Where KdT = Binding association for target T

Selectivity of tissue expression

 T = Expression in different tissue 

 ρ_i = Probability of an expression value

ρ_((T))=E_T /(∑_i▒〖E_T〗_i )

 Where ET = Expression value in tissue T

• It appears that there are no resolution cut-offs used for structures that go through fpocket analysis. Please can you explain if that is the case and why you feel this is appropriate, if so? An EM structure of >6 Angstroms is going to provide unreliable results in this context, for example.

Answer: We thank the reviewer for his suggestion, as of now, it is true that no cut-offs are put in place but this will certainly be implemented in the next release of the database. In the meantime, the information on the resolution of the structure is captured on a different tab and can be used to manually remove any binding pocket result coming from such low resolution structures. (This has been captured as a enhancement request on the GitHub page: https://github.com/sdecesco/targetDB/issues/10)

• What is the sustainability plan for targetDB? Since databases change all the time, how will you ensure that this platform is still usable even in 6 months’ time?

Answer: As of now two avenues are open, we are committed to update the database bi-annually for the foreseeable future, and are also working on a separate version which would collect data live from various API of these datasources. The disadvantage of the live version would be that it would be slower and would more easily break down if any of the used datasource modify their outputs in a significant manner. 

Moreover, one of the main reason to make the entire source code available is to allow anyone with enough experience in python to modulate and/or fix TargetDB in the future. 

• It is highly unusual to use ellipses in articles. Please remove these and use ‘etc.’ (for example) instead.

Answer: Ellipses have been removed in the revised manuscript

• It would be good to have the article carefully proof read for English before it is finally accepted.

Answer: We thank the reviewer for the advice and the manuscript has now been carefully proofread by native English speakers.

Overall, a nice piece of work that I’m sure many will want to use.

Answer: We thank the reviewer again for their kind words.

Reviewer #3

The authors construct a database application with a graphical interface that aggregates multiple disparate sources of evidence for novel drug target evaluation. They consider measures of druggability, structure, chemistry, biology, disease association, genetic association, general information, and safety when constructing their application. As output they produce different summaries of this evidence - either a summary for a single target, multiple targets, or a graphical summary (aka a spider plot). Furthermore, the authors provide an optimization approach (powered by a machine learning method) to allow users to weight multiple classes of evidence based on the users target validation needs and interests.

Major comments -

Overall the authors provide a potentially useful tool to aid structural biologists, chemists, and assay developers for target prioritization. That being said, I have the following comments:

1. How reproducible are the results generated by this - many of the databases that evidence is being pulled from may change over time. Is there some way to version the results, or to construct queries that refer to a specific version of a public database?

Answer: The current model is that the database behind TargetDB is generated at a certain date and time by pulling data from specific versions of databases (which are specified in the supporting information and/or fields in the database itself) or through the API of the sources at that same date (+/- few days as the creation of the TargetDB database takes a few days to complete) 

2. The user defined weighting is an interesting idea - I would like to see the authors discuss potential pitfalls of this - e.g. over optimizing for targets with interesting structural biology/chemistry as opposed to targets that will actually lead to therapies or deeper insights into the disease biology.

Answer: This is an interesting and important point raised by the reviewer. Providing this tool of course does not guarantee that it will always be used in a way that leads to therapies. We have added a few words of caution regarding the use of these weights in target prioritization: 

“As note of caution, while a user might decide to over-prioritize areas such as structural biology and chemistry it is by no means a guarantee that this will lead to a target with potential therapeutic applications. This facility also enables users to tailor their searches to highlight targets where there is an opportunity for their research expertise to make an impact; picking targets for which there is a lack structural information, lack of chemistry or even lack of biology, for example On the other hand, if a target is over-prioritized for strong disease and genetic links, it may be very difficult to develop safe and effective therapeutics due to low safety and structural druggability scores.”

Minor comments -

1. use of ellipses is distracting (Abstract, Introduction Paragraph 1).

Answer: These have now been removed from the revised manuscript (see reviewer #2 comments)

2. “Percentage of threes predicting” - spelling/grammar throughout should be double checked.

Answer: Spelling and grammar has now been checked throughout the document

3. SBDD acronym in Table 1 is not defined.

Answer: Definition of the acronym has now been added to the text

---

## [Editor Report · Decision Letter 1]

29 Jul 2020

TargetDB: A target information aggregation tool and tractability predictor

PONE-D-20-11085R1

Dear Dr. De Cesco,

We’re pleased to inform you that your manuscript has been judged scientifically suitable for publication and will be formally accepted for publication once it meets all outstanding technical requirements.

Kind regards,

Joseph J Barchi

Academic Editor

PLOS ONE
---

## [Editor Report · Acceptance letter]

3 Aug 2020

PONE-D-20-11085R1 

TargetDB: A target information aggregation tool and tractability predictor 

Dear Dr. De Cesco:

I'm pleased to inform you that your manuscript has been deemed suitable for publication in PLOS ONE. Congratulations! Your manuscript is now with our production department. 

Kind regards, 

on behalf of

Dr. Joseph J Barchi 

Academic Editor

PLOS ONE